# Cutaneous disseminated sporotrichosis associated with diabetes: A case report and literature review

**Xiujiao Xia**  **\*, Huilin Zhi, Zehu Liu\***

Department of Dermatology, Hangzhou Third People's Hospital, Affiliated Hangzhou Dermatology Hospital, Zhejiang University School of Medicine, Hangzhou, China

\* 804534095@qq.com (XX); zehuliu@yahoo.com (ZL)

## Abstract

### Background

Cutaneous disseminated sporotrichosis (CDS), also called hematogenous sporotrichosis, is a rare condition that usually affects immunocompromised patients. The current work presents the case of a woman with diabetes mellitus associated with CDS.

### Case presentation

A 59-year-old woman with diabetes mellitus presented with a two-year history of ulcerated rashes on the left ankle and both sides of the jaw. Physical examination revealed three annular areas of erythematous and raised plaque with an ulcer over the left ankle and both sides of the jaw. Based on laboratory findings, elevated blood glucose concentration and decreased white cell count were observed. *Sporothrix globosa* was identified in the mycological culture of biopsied tissue from the three lesions and this was confirmed by DNA sequencing. The skin lesions healed after two-month itraconazole therapy.

### Conclusions

Diabetes is a risk factor for disseminated sporotrichosis, which may be induced by hematogeneous spread, repeated inoculation, or autoinoculation. This study raises awareness among clinicians, with regard to the notion that people with possibly altered immune function are potentially vulnerable to severe clinical forms of sporotrichosis.

**Data Availability Statement:** All data generated or analysed during this study are included in this published article.

**Funding:** This work was supported by a grant from the Hangzhou Science and Technology Bureau

## Author summary

Sporotrichosis is one of the unspecified deep mycoses that match the neglected tropical diseases (NTDs) selection criteria. It is caused by *Sporothrix*, a fungus that usually results in zoonotic fungal diseases and sapronosis that are endemic in temperate regions. Sporotrichosis is clinically presented in two main forms, which are lymphocutanous sporotrichosis (LC) and fixed sporotrichosis (F), especially in immunocompetent patients.

(http://kj.hangzhou.gov.cn) grant 202004A17 to ZHL. The funder had no role in study design, data collection and analysis, decision to publish, or preparation of the manuscript.

**Competing interests:** The authors have declared that no competing interests exist.

However, severe clinical versions, such as cutaneous disseminated, mucosal, and extracutaneous forms may present in immunocompromised hosts. This study presents a case of cutaneous disseminated sporotrichosis in a patient with diabetes mellitus and information based on the review of relevant literature. Clinicians should be on the lookout for severe clinical forms of sporotrichosis in all patients with possibly altered immune function.

## Introduction

Sporotrichosis is a subacute or chronic implantation (formerly subcutaneous) fungal infection, related to dimorphic fungi that fall within the *Sporothrix schenckii* complex [1–4]. In humans, the infection is prompted by traumatic inoculation from contaminated sources, such as wood, thorns, and splinters. This explains why it is known as "gardener's disease." The infection can also be transmitted through scratches and bites from infected cats [5].

Based on the location of the lesions, sporotrichosis can be classified into cutaneous, mucosal, and extracutaneous forms. The cutaneous form is more common [6]. Clinically, the infection is mainly presented as lymphocutanous sporotrichosis (LC) or fixed sporotrichosis (F). Cutaneous disseminated sporotrichosis (CDS) has also been reported, although it is less common [7]. CDS is also called hematogenous sporotrichosis and even though it is rare, more incidences have been reported in immunocompromised patients [4]. CDS is characterized by multiple skin lesions at non-adjacent sites, without extracutaneous involvement. This report presents the case of a woman with diabetes mellitus, who suffered from a *S. globosa* infection that disseminated to the jaw and ankle.

## Case report

In March 2020, a 59-year-old woman with diabetes mellitus presented with a two-year history of ulcerated rashes on the left ankle and both sides of the jaw. Three years prior to the presentation, the patient was diagnosed with type 2 diabetes at a local hospital but had not received standard treatment. The patient worked at a construction site but denied a history of skin trauma. The skin lesions were slightly itchy and sometimes painful. The patient had been assessed at another clinic, where she received surgical treatment for a suspected cutaneous tumor. Physical examination revealed three annular areas of erythematous and raised plaque with an ulcer over the left ankle and both sides of the jaw (Fig 1A–1C).

The fasting blood glucose concentration was 10.41 mmol/L (reference range: 3.90–6.10 mmol/L) while the white blood cell count was 3800 cells/mm$^3$ (reference range: 4000 to 10000 cells/mm$^3$), with 47.7% neutrophils and 30.2% lymphocytes. Liver and renal function tests were done, and the results were within normal ranges. HIV serology was negative. Direct microscopy of exudate smears in either 10% of potassium hydroxide solution (KOH) or fluorescence staining displayed slender hyphae (Fig 2A). Biopsies were taken from skin lesions on the left chin and left ankle, before being immediately sent for histopathological examination and fungal culture. Both lesions showed similar histopathological responses, with pseudoepitheliomatous hyperplasia in the epidermis; neutrophil abscess in the superficial dermis, diffuse lymphocytes, histiocytes, plasma cells, and a few neutrophils infiltrated in the dermis; and multinucleated giant cells.

Periodic acid-Schiff stain revealed dark red spores in multinucleated giant cells (Fig 2B). The mycological culture of exudate from the three lesions and biopsied tissues from the two lesions were performed on Sabouraud dextrose agar that contained 0.05% chloramphenicol, at 25˚C. The growth of creamy, brownish-beige colonies that are typical of *Sporothrix* spp.

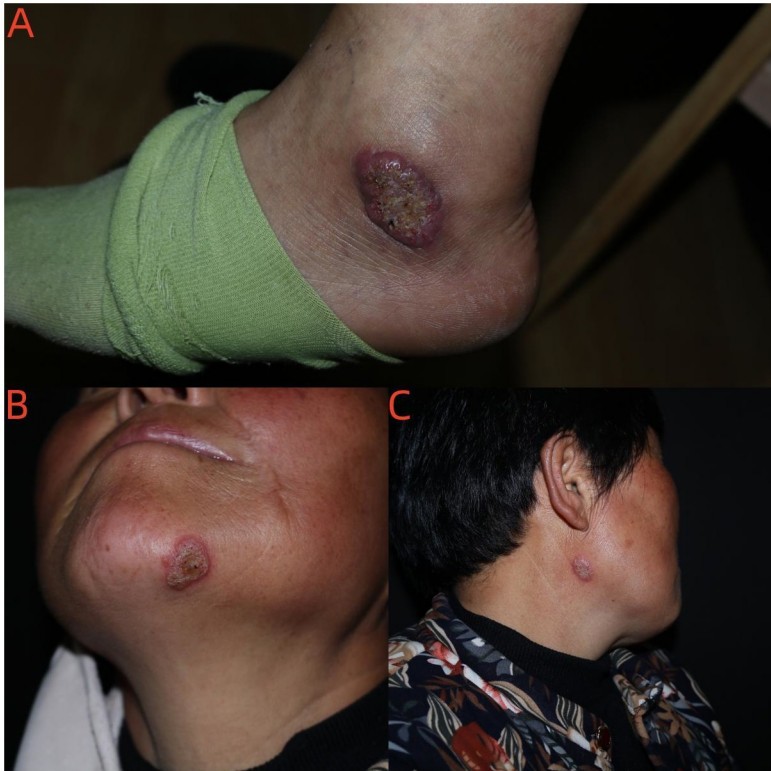

**Fig 1. Plaques with ulcers located on the ankle (A) and both sides of the lower jaw (B, C).**

was noted after incubation (Fig 2C). All of the samples developed similar colonies on SDA slant media. Microculture showed thin septate hyphae with conidia in a dense arrangement at its extremities. The fungal culture was confirmed to be *S. globosa* by sequencing the ribosomal region ITS1-ITS2 of the rDNA (GenBank accession no. OP614927). Conclusively, the patient was diagnosed with cutaneous disseminated sporotrichosis caused by *S. globosa* infection.

Antifungal susceptibility testing using commerical microdilution plate (Trek Diagnostic Systems, Ltd., East Grinstead, United Kingdom) was carried out based on the manufacturer's instructions. The minimum inhibitory concentration (MIC) values were as follows: anidulafungin (0.25 μg/ml), micafungin (0.5 μg/ml), fluconazole (>256 μg/ml), itraconazole (0.5 μg/ml), voriconazole (>8 μg/ml), caspofungin (1 μg/ml), 5-flucytosine (4.0 μg/ml), posaconazole (1.0 μg/ml) and amphotericin B (<0.12 μg/ml).

Considering the clinical epidemiological diagnosis on the patient, therapy with itraconazole was initiated at 400 mg/day, for two months. During treatment, liver and kidney function were normal; blood glucose concentration varied between 9.85 mmol/l and 12.57 mmol/l; and the WBC count varied from 3400 cells/mm$^3$ to 3700 cells/mm$^3$. The patient declined diabetes treatment and tests for identifying the cause of the low count of white blood cells. On the 40[th] day, Leucogen Tablets (60 mg/day) were administered to the patient to raise the white blood cell count. Finally, after a two-month course of treatment, the lesions resolved, and the WBC count rose to 4100 cells/mm$^3$. There was no recurrence after three years of follow-up (Fig 3A and 3B) and the patient is currently being treated for diabetes at a local hospital.

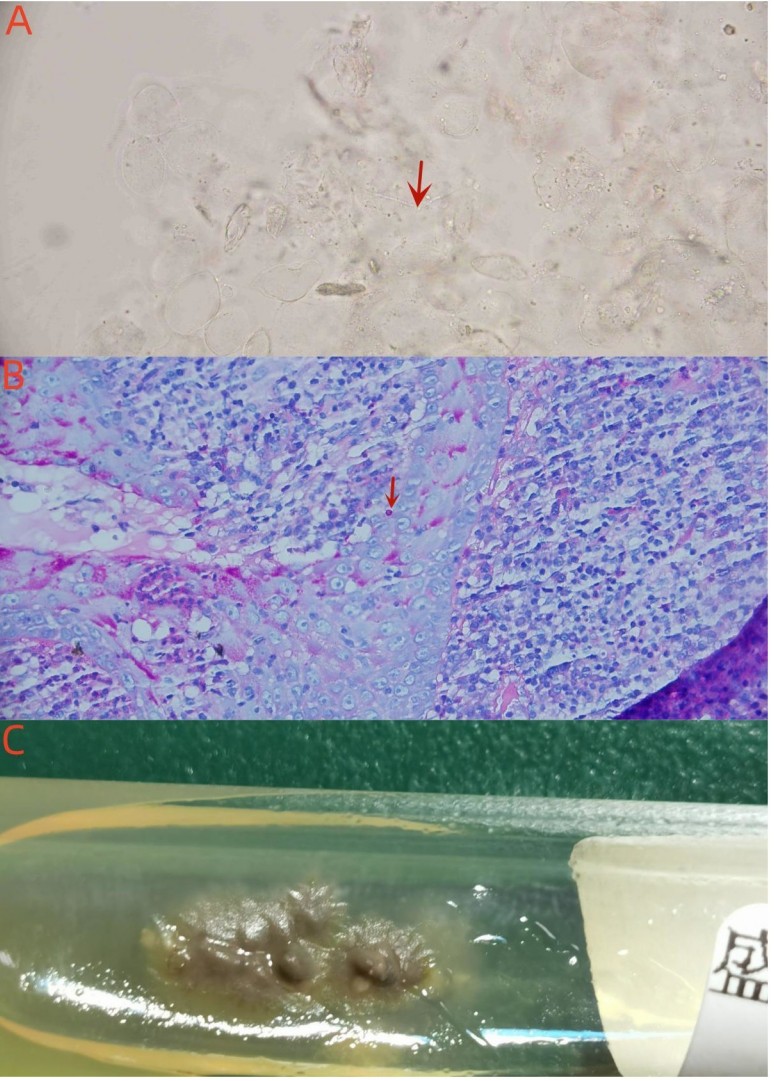

**Fig 2.** Direct microscopy of exudate smears showing slender hyphae (A, 10% KOH × 400). Periodic acid-Schiff staining showing fungal spores in multinucleated giant cells (B, × 1000). *S. globosa* colonies on Sabouraud dextrose agar at 25˚C on day 9 (C).

## Methods

### Ethics statement

This study was approved by the ethics committee of Hangzhou Third People's Hospital and the study participant was informed about the study procedures and written informed consent was obtained.

A systematic literature search cases of sporotrichosis in patients with diabetes has been herein conducted on March 2023, by sourcing both National Library of Medicine (NLM) resources through PubMed, and Scopus and Web of Science, using the following keywords: ("Sporotrichosis") AND ("diabetes" OR "case report). We first screened the resources, read the titles and abstracts of the articles, and removed the duplicate content, and finally selected 33 items. Then we analyzed the articles according to the inclusion criteria, as topic on

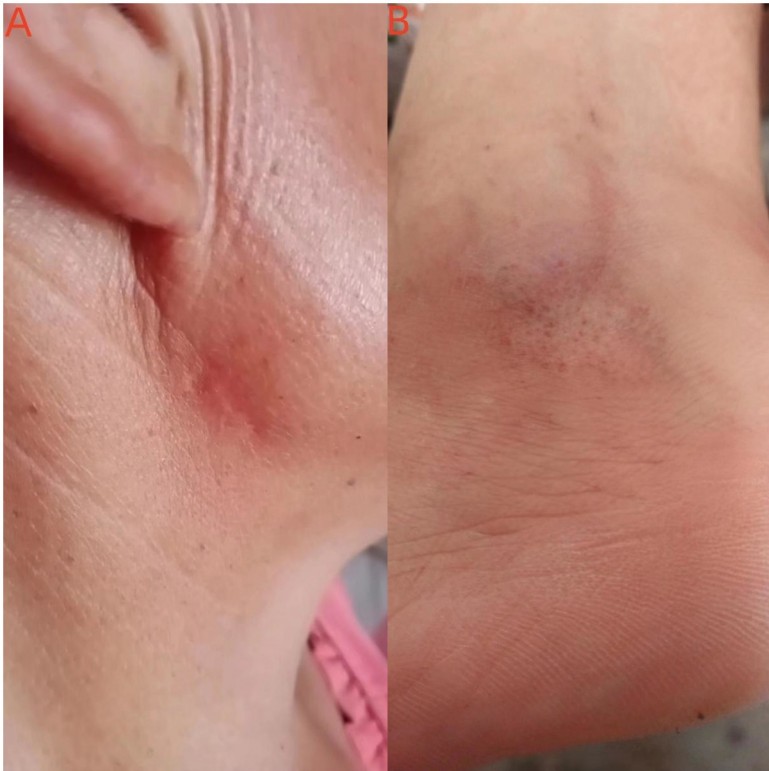

**Fig 3. The skin lesions (A, the left ankle; B, the right jaw) were disappeared at 3-year follow-up.**

sporotrichosis associated with diabetes, English language studies, case reports or research articles, and exclusion criteria, as no risk factor studies or accompanied by other major underlying diseases, resulting in the selection of 16 items. The article details included in the review are reported in Table 1.

## Results

In total, 14 articles with 16 patients were identified according to a literature search. Overall, the number of reported cases remains single digit. The detailed characteristics of the 16 cases of sporotrichosis in patients with diabetes are presented in Table 1. There were 14 male and 2 female patients, with median age was 59 (range, 23–89) years. Of the 16 patients, 9 presented with cutaneous form and 7 presented with extracutaneous form. Among the 15 patients who received antifungal therapy, 1 died and the others had a good prognosis.

## Discussion

Sporotrichosis is a subacute/chronic fungal disease that is caused by the dimorphic fungus of the genus *Sporothrix*. This fungal disease can affect both humans and livestock [7]. At present, the consensus is that the S. *schenckii* complex is composed of S. *brasiliensis*, S. *Schenckii*, S. *globosa*, S. *mexicana*, and S. *luiriei* [8–11]. S. *globosa* and S. *schenckii* are globally distributed, with the former serving as the primary pathogen for human sporotrichosis in China [12]. S. *globosa* is found in decomposing plants, soil, and mosses in various parts of the world. It's optimum conditions for growth are 22˚C and 27˚C (71˚ F and 80˚ F) and 90% humidity. S. *globosa* requires cellulose-rich soil, and a pH level between 3.5 and 9.4 [13].

**Table 1. Cases of sporotrichosis in patients with diabetes as reported in scientific literature until March 2023.**

| Case | Country | Age (years), sex | Presentation | Treatment/Course or total dose | Outcome | Reference |
|------|---------|------------------|--------------|-------------------------------|---------|-----------|
| 1 | Spain | 72, male | CDS | ITR/5 months | Recovered | 16 |
| 2 | Peru | 67, male | F | SSKI/2 months | Recovered | 58 |
| 3 | Mexico | 78, male | Fungemia | Untreated/NA | Died | 59 |
| 4 | Brazil | 55, female | Disseminated | AmB/2.5 g | Recovered | 60 |
| 5 | Japan | 53, male | Pulmonary | AmB/Unknown | Unknown | 61 |
| 6 | America | 23, male | Pulmonary | AmB/100 mg | Unknown | 62 |
| 7 | China | 59, male | LC | ITR/10 months | Recovered | 63 |
| 8 | China | 60, male | LC | ITR/6 months | Recovered | 63 |
| 9 | Korean | 70, male | F | ITR/2 months | Improved | 64 |
| 10 | America | 67, male | LC | ITR/6 weeks | Recovered | 65 |
| 11 | America | 27, male | Ocular | AmB/215 mg | Recovered | 66 |
| 12 | Italy | 64, male | LC | ITR/8 months | Recovered | 27 |
| 13 | Mexico | 40, male | LC | ITR + cryosurgery session/6 weeks | Recovered | 67 |
| 14 | Mexico | 62, female | F | ITR + cryosurgery session/8 weeks | Recovered | 67 |
| 15 | America | 89, male | Disseminated | Unknown | Unknown | 68 |
| 16 | America | 58, male | Fungemia | ITR/3 months | Recovered | 69 |

CDS, cutaneous disseminated sporotrichosis; F, fixed sporotrichosis; LC, lymphocutanous sporotrichosis; ITR, itraconazole; SSKI, saturated solution of potassium iodide; AmB, amphotericin B; KCZ, ketoconazole

The infection route is predominantly by traumatic inoculation with contaminated material. However, most patients forget their history of trauma because wounds are typically mild and usually occur a few weeks earlier. Approximately 10 to 62% of the patients recall the trauma related to infection [14–15]. In this case, although the patient denied a history of trauma, the patient was categorized with susceptible populations due to her involvement in construction work for a long time. There is a probability that inconscient traumatic inoculation might have taken place.

Sporotrichosis manifests as a papule or nodule at the trauma site, within a couple of days after traumatic inoculation. After penetrating the skin, the fungus transforms into the yeast form and may remain localized in the subcutaneous tissue or extend along adjacent lymphatic vessels, constituting the fixed or the lymphocutaneous form, respectively [4]. The fungus may be transmitted via a hematogenous route, although this is rare and is characterized by a cutaneous disseminated form [16].

Most disseminated, visceral, and fungemia forms of sporotrichosis are associated with immunosuppression and studies report less than 5% [3,4,17]. De Beurmann et al were the first to describe CDS, the third form of cutaneous sporotrichosis, which is a rare variant with multiple lesions without extracutaneous involvement. In their reports, these researchers suggested that the fungus acted as an opportunist [18]. Since then, single or serial cases of CDS were brought forward, only reported in up to 8% of the total cases of sporotrichosis [3,4]. However, there are some exceptions. For example, a report from Mexico showed 24 out of 174 cases of CDS (13,8%) [19]. Generally, the frequency of CDS in our department is not high. From January 2013 to August 2022, a total of 71 cases of cutaneous sporotrichosis were identified, of which only one case was CDS (1/71, 1.41%) [20].

The most frequent predisposing factor of CDS was HIV-AIDS [3,21–24], usually occurring in patients in stage C of AIDS (CD4+ count <200 cells/ mm$^3$) [3,17]. Second, CDS can be caused by uncontrolled diabetes [3,4,25]. Overall, the condition may be attributed to

immunological deterioration related to cellular response, especially macrophage dysfunction [3,4,26]. Alterations in skin trophism, as well as in specific and non-specific local defenses are important predisposing factors that make diabetic patients more prone to infections [27]. Our patient had uncontrolled diabetes and leukopenia, a scenario that possibly exposed her to CDS. However, in the case reported in this study, the relationship between diabetes and leukopenia remains unclear. Saygin C *et al.* reported an adult type 1 diabetes case with neutropenia associated with anti-neutrophil cytoplasmic antibodies [28]. Another study found that mild but significant neutropenia preceded and accompanied type I diabetes [29]. More research showed an inverse relationship between white blood cell (WBC) count and insulin tolerance, suggesting that leukocytosis was associated with the development and possibly, the progression of diabetes mellitus [30]. In addition, CDS also has been reported in immunocompromised patients with chronic alcoholism [19,31–33], hematologic cancer (leukemia and lymphomas) [34–36], and malnutrition [3,4]. Those who have undergone steroid treatment [2,3,37], and organ transplantation [38,39] have also been classified as relatively more vulnerable to CDS.

It's also important to note that there are exceptional cases where immunocompetent patients are affected by CDS [17,40–43]. This suggests that there are multiple factors from the fungus and host that contribute to the clinical evolution of sporotrichosis to benign or severe disease [7]. Different species of the genus *Sporothrix* have varying grades of virulence, mainly based on the role of melanine, glycoproteins, and other cell wall components as markers of virulence. These grades of virulence are based on the capacity of the fungi to evade immune recognition [44] and are recognized by human mononuclear cells [45], inducing an immune response [46]. An immunosuppressant status is an essential factor for cutaneous-disseminated and disseminated sporotrichosis [47]. However, Zhang *et al.* found a 10-bp deletion in the ribosomal NTS region of a strain that was isolated from a CDS patient versus control strains that were obtained from fixed cutaneous sporotrichosis [48]. These results suggested the existence of a more virulent strain that may produce cutaneous-disseminated and disseminated sporotrichosis [3,4,17].

Although there are distinct pathogenesis characteristics for each situation, determining whether the multiple cutaneous sporotrichosis lesions are due to repeated inoculations, hematogenous dissemination, or even self-inoculation is a challenge [4]. Our patient's occupation probably offered relatively enough opportunities for environmental exposure to pathogenic *Sporothrix* spp. Moreover, the fact that she had diabetes mellitus and leukopenia makes it even more difficult to ascertain the cause of the spread of skin lesions. One study suggested that repeated inoculation was more likely to be predominantly responsible for the clinical presentation in CDS patients [49]. The manifestations of CDS were mainly nodules, papules, necrotic lesions, ulcers, and verrucous plaques located in different sites of the body [3,4,17,19,24,50–52]. It is unclear whether clinical manifestations are related to the route of transmission or not.

The standard criterion for sporotrichosis diagnosis involves isolating the etiologic agent [4]. Direct examination and staining are not useful for diagnosing cutaneous-lymphatic and cutaneous-fixed sporotrichosis, since yeasts are observed only in a low percentage (5%–10%) [47]. However, in our recent series of studies, we found that the direct microscopic positive rate of cutaneous sporotrichosis was as high as 40.91%, with hyphae, budding yeast or spores visible under the microscope [20]. Especially in cases of immunocompromised patients, large clusters of yeast are observed, similar to feline sporotrichosis [53,54]. The presence of slender hyphae under direct microscopic may be attributed to low body surface temperature in the sites of the patient's open skin lesions. So, in CDS patients, direct microscopy is an important diagnostic procedure, considering the high fungal burden in tissues of immunocompromised patients [21,55]. On the other hand, the presence of numerous yeast cells or hyphae upon conducting

direct mycological examination and histopathology may be a red flag for a compromised immune system or a highly invasive pathogen fungus.

Generally, itraconazole is effective against any form of sporotrichosis in immunocompromised patients. The dosage varies from 200 mg daily or twice daily, depending on the clinical presentation [4]. However, the treatment guidelines stipulate that amphotericin B, preferably lipidic at 3–5 mg/kg/day, should be administered for more severe forms of sporotrichosis [55]. If the deoxycholate form is used, the recommended doses are 0.7 to 1 mg/kg/day while the treatment duration varies depending on the response and side effects (mainly renal damage) [56,57]. In this case, the patient adequately responded to treatment with itraconazole. For CDS, the choice of treatment should be focused on the route of transmission, with amphotericin B being the first option in the case of hematogenous spread, while itraconazole is considered in cases of self-inoculation or repeated inoculations.

The risk of severe sporotrichosis primarily depends on the immune status of the patients, and the risk is significantly higher in diabetics, although lower than in AIDS patients, compared to immunocompetent patients. Table 1 indicates a male predominance, with the cutaneous and extracutaneous forms accounting for nine and seven cases, respectively [16,27,58–69]. Itraconazole and amphotericin B (AmB) were the most common antifungal regimens that were prescribed in patients for cutaneous and extracutaneous forms, respectively. This suggests that diabetes is a risk factor for severe sporotrichosis, and clinicians should be aware of the potential for severe clinical forms of sporotrichosis in all people with possibly compromised immune function. Invasive fungal disease is a life-threatening complication in type 2 diabetes mellitus patients. *C. albicans*, *C. neoformans*, and *A. fumigatus* are the leading agents. Prolonged hyperglycemia results in unfavorable outcomes. Correction of anemia and hypoalbuminemia might improve prognosis [70]. Similarly, for the treatment of severe sporotrichosis, the control of other underlying diseases such as diabetes is a necessary measure.

## Acknowledgments

The authors wish to thank the patient for her willingness to share her experience by providing informed consent for this case report.

## Author Contributions

**Conceptualization:** Xiujiao Xia.

**Data curation:** Huilin Zhi.

**Formal analysis:** Xiujiao Xia, Huilin Zhi.

**Funding acquisition:** Zehu Liu.

**Investigation:** Xiujiao Xia, Huilin Zhi.

**Methodology:** Xiujiao Xia, Huilin Zhi.

**Project administration:** Xiujiao Xia, Zehu Liu.

**Resources:** Xiujiao Xia, Huilin Zhi.

**Supervision:** Xiujiao Xia, Zehu Liu.

**Validation:** Xiujiao Xia, Zehu Liu.

**Visualization:** Xiujiao Xia, Huilin Zhi.

**Writing – original draft:** Xiujiao Xia.

**Writing – review & editing:** Xiujiao Xia, Huilin Zhi, Zehu Liu.

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
