## [Decision Letter · Decision Letter 0]

19 Jun 2023

Dear Mr Xia,

Thank you very much for submitting your manuscript "Cutaneous disseminated sporotrichosis associated with diabetes: A case report and literature review" for consideration at PLOS Neglected Tropical Diseases. As with all papers reviewed by the journal, your manuscript was reviewed by members of the editorial board and by several independent reviewers. The reviewers appreciated the attention to an important topic. Based on the reviews, we are likely to accept this manuscript for publication, providing that you modify the manuscript according to the review recommendations. 

Sincerely,

Joshua Nosanchuk, MD

Section Editor

Reviewer's Responses to Questions

**Key Review Criteria Required for Acceptance?**

**Methods**

-Are the objectives of the study clearly articulated with a clear testable hypothesis stated?

-Is the study design appropriate to address the stated objectives?

-Is the population clearly described and appropriate for the hypothesis being tested?

-Is the sample size sufficient to ensure adequate power to address the hypothesis being tested?

-Were correct statistical analysis used to support conclusions?

-Are there concerns about ethical or regulatory requirements being met?

Reviewer #1: The structure and methodology of the case is adequate

Reviewer #2: Actually, the article format of this MS is case report instead of research article, so I suggest that this submission is more suited as case report format. 

Besides, could the author offer the figure of direct microscopy, because it was stated that direct microscopy displayed slender hyphae. Moreover, it was stated that the patient healed after two months' therapy, but there are not any figures after healed.

Reviewer #3: The draft manuscript “Disseminated cutaneous sporotrichosis (CDS) refers to the case of a woman with diabetes mellitus associated with CDS”. Based on the laboratory findings, there was an increase in blood glucose and a decrease in the leukocyte count, standard for mycological diagnosis and confirmed by DNA sequencing. The study adequately described antifungal therapy. The paper can be accepted, but be interesting to adapt it to the suggestions below.

**Results**

-Does the analysis presented match the analysis plan?

-Are the results clearly and completely presented?

-Are the figures (Tables, Images) of sufficient quality for clarity?

Reviewer #1: The results of the case are adequate

Reviewer #2: The MS reviewed the literatures of CDS, but only cited the table in the results of male predominance. The table could review the data of underling disease and the duration of treatment. 

Besides, the MS stated that the risk factor for CDS is diabetes but did not show the results of high-throughput genome sequencing or test for HIV.

Reviewer #3: The authors reported the results of the leukogram as its monitoring during the treatment, thus verifying the importance of the procedure but not the information on the red blood cells. In these patients, anemia and hypoalbuminemia may occur in parallel. Add the information, if possible, and insert the reference below.

Lao et al. (2020) found that invasive fungal disease in patients with type 2 diabetes mellitus does not have a good prognosis, with complications potentially fatal, as prolonged hyperglycemia results in unfavorable outcomes. In these patients, anemia and hypoalbuminemia occur in association, and their correction helps to improve the prognosis.

Lao et al. (2020) mention the relationship between diabetes and leukopenia, even if they do not observe its significance. The fact that the clinical improvement of the reported patient is favorable with the improvement of his leukogram suggests future investigations.

The authors mention that the patient had uncontrolled diabetes. It is relevant to report whether it is type 1 or type 2 diabetes and whether the treatment is adequate for your underlying disease.

**Conclusions**

-Are the conclusions supported by the data presented?

-Are the limitations of analysis clearly described?

-Do the authors discuss how these data can be helpful to advance our understanding of the topic under study?

-Is public health relevance addressed?

Reviewer #1: The conclusions reached are correct.

Reviewer #2: The MS stated that the risk factor for CDS is diabetes but did not show the results of high-throughput genome sequencing or test for HIV.

Reviewer #3: The study would be more impactful if the authors added other studies of diabetes and fungal infections to transpose this information into sporotrichosis since is still obscure.

**Editorial and Data Presentation Modifications?**

Reviewer #1: No

Reviewer #2: (No Response)

Reviewer #3: The study would be more impactful if the authors added other studies of diabetes and fungal infections to transpose this information into sporotrichosis since is still obscure.

**Summary and General Comments**

Reviewer #1: It is an interesting case that emphasizes how cases of sporotrichosis can spread through the skin.

The only thing that I suggest to the authors that they add in the discussion is that these cases usually present little yeast presence, are similar to lymphangitic cases and that they behave differently when there is immunosuppression because the number of yeasts is greater

Reviewer #2: (No Response)

Reviewer #3: Suggestion of reference

Lao M, Li C, Li J, Chen D, Ding M, Gong Y. Opportunistic invasive fungal disease in patients with type 2 diabetes mellitus from Southern China: Clinical features and associated factors. J Diabetes Investig. 2020 May;11(3):731-744. doi: 10.1111/jdi.13183. Epub 2019 Dec 20. PMID: 31758642; PMCID: PMC7232281.

Format the references according to the journal's rules.

Reference 42 is incorrect.

PLOS authors have the option to publish the peer review history of their article (what does this mean?). If published, this will include your full peer review and any attached files.

Reviewer #1: Yes: Alexandro Bonifaz

Reviewer #2: No

Reviewer #3: Yes: Roseli Freitas-Xavier

Figure Files:

Data Requirements:

Reproducibility:

References

---

## [Decision Letter · Decision Letter 1]

21 Aug 2023

Dear Mr Xia,

Thank you very much for your thoughtful revision of your manuscript "Cutaneous disseminated sporotrichosis associated with diabetes: A case report and literature review" for consideration at PLOS Neglected Tropical Diseases. As with all papers reviewed by the journal, your manuscript was reviewed by members of the editorial board and by several independent reviewers. The reviewers appreciated the attention to an important topic. Based on the reviews, we are likely to accept this manuscript for publication, providing that you modify the manuscript according to the review recommendations. 

Reviewer comments: 

-Thank you for the revision. 

-It is hard to find the slender hyphae in Figure 2A. And as slender hyphae are not commonly seen in direct smear, the author should discuss this finding. 

-It is not clear if all lesions were culture positive. Please address this point.

Sincerely,

Joshua Nosanchuk, MD

Section Editor

Figure Files:

Data Requirements:

Reproducibility:

References

---

## [Editor Report · Decision Letter 2]

7 Sep 2023

Dear Mr Xia,

Thank you for your thoughtful revisions! We are pleased to inform you that your manuscript 'Cutaneous disseminated sporotrichosis associated with diabetes: A case report and literature review' has been provisionally accepted for publication in PLOS Neglected Tropical Diseases.

Best regards,

Joshua Nosanchuk, MD

Section Editor

---

## [Editor Report · Acceptance letter]

13 Sep 2023

Dear Mr Xia,

We are delighted to inform you that your manuscript, "Cutaneous disseminated sporotrichosis associated with diabetes: A case report and literature review," has been formally accepted for publication in PLOS Neglected Tropical Diseases.

Best regards,

Shaden Kamhawi

co-Editor-in-Chief

Paul Brindley

co-Editor-in-Chief
